# Analysis of the Effectiveness of Green Waste Composting under Hyperbaric Conditions

**Jerzy Bieniek [1], Błażej Gaze [1,*], Bernard Knutel [1], Krzysztof Rać [1] and Sara Góraj [2]**

1 Institute of Agricultural Engineering, Wroclaw University of Environmental and Life Sciences, 51-630 Wroclaw, Poland; jerzy.bieniek@upwr.edu.pl (J.B.); bernard.knutel@upwr.edu.pl (B.K.); krzysztof.rac@upwr.edu.pl (K.R.)

2 Faculty of Life Sciences and Technology, Wroclaw University of Environmental and Life Sciences, 50-363 Wroclaw, Poland; 118406@student.upwr.edu.pl

* Correspondence: blazej.gaze@upwr.edu.pl; Tel.: +48-71-320-57-15

**Abstract:** Increasing global population growth has a significant impact on waste production. The European Union (EU) focuses on waste recycling, biological treatment, and reuse. In the case of biodegradable waste, a significant problem is the long process of material decomposition so that the product meets the requirements of national regulations and EU directives. The search for a way to accelerate this process is still ongoing. This study presents the composting process of green waste under hyperbaric conditions. Eight experiments, four with air exchange frequency $t_{AE} = 4$ h and four experiments with $t_{AE} = 8$ h were established. The experiments were conducted in four variants: 0 (atmospheric pressure) and overpressures 50, 100, and 200 kPa. They were carried out on the same input material characterized by the initial moisture content of 60% and a mass of 2000 g. During the composting of green waste, all parameters of the obtained product (moisture content, pH, loss on ignition (LOI), C:N ratio, nutrient content (P, K), and the respiratory activity of microorganisms ($AT_4$)) were also evaluated. The most significant weight loss of the composted material was observed in the variant of an overpressure of 200 kPa ($t_{AE} = 4$ h). The compost weight in relation to the original material decreased by 23.7%. The highest organic matter removal efficiency was obtained for the overpressure variants of 50 and 100 kPa ($t_{AE} = 4$ h).

**Keywords:** composting; green waste; hyperbaric conditions; effectiveness

## 1. Introduction

Currently, the world is struggling with the problem of generating a massive amount of municipal waste. The aftermath of this problem is the phenomena of illegal garbage dumps or plastic stains in the Pacific Ocean [1]. Globalization contributed to the aggravation of the above-mentioned problems and the rapid development of new technologies, most of which did not consider the principles of sustainable development in their program. Thus, such vast waste masses began to harm the environment.

The worst possible solution is landfilling, which harms the environment. Unprocessed waste taking up the environmental space that could be managed more efficiently, makes the place an unfriendly point on the map for humans and animals. Above all, inanimate nature suffers from dumping waste. Leachate penetrating the layers of the landfill goes to the soil and then to the groundwater, contaminating it. It also lowers the quality of the air into which all waste vapors and byproducts of biochemical reactions taking place in the waste mass evaporate (i.e., carbon dioxide, hydrogen sulfide, aldehydes, organic acids, or methane), which can influence and deepen the greenhouse effect, acid rainfall, and the ozone hole [2–4].

The negative aspect of waste management is nonselective waste collection. Some of the mixed waste belongs to the bio fraction. The presence of biowaste rich in organic substances increases waste's moisture, weight, and volume [5]. Such conditions favor biochemical changes.

The impact mentioned above motivated the European Union in 1975 to introduce Directive 74/442/EEC presenting a strategy in waste management [6]. The current rules are defined in Directive 2008/98/EC of the European Parliament and the Council [7]. It establishes a waste management hierarchy and aims to prevent waste generation, and encourage its reuse, recycling, recovery, and the disposal of hazardous waste. Additionally, several regulations have been developed to closely regulate the classification and handling of waste. These rights came into force in Poland on its accession to the European Union in 2004. Previously, Polish waste management was regulated by the Act of 27 April 2001 on Waste (Journal of Laws No. 62, item 628), the provisions of which mostly coincided with those implemented regulations of the European Union [8,9].

In the entire European Union, 37 million tons of biowaste are generated annually [10], of which in Poland, according to the data of the Central Statistical Office of 2017, it is over 890 thousand tons [11]. Biowaste is defined following the Act of 14 December 2012 on Waste as biodegradable waste from gardens and parks, food and kitchen waste from households, gastronomy, mass caterers, retail units, and similar waste from establishments producing or marketing food [12]. They are another problem for waste management, as they constitute a large percentage of municipal waste, and if not properly segregated, they contaminate glass, paper, and plastic fractions.

Treatment of biowaste in the territory of the Republic of Poland is specified in the Act of 19 July 2019 (Journal of Laws, item 1579). Separate waste collection is recommended, in which the breakdown includes the biowaste fraction. The commune is responsible for collecting such a fraction, enabling the transfer of waste to a selective collection point, and then processing it following the hierarchy of waste management methods and the principle of proximity. The amount and type of processed waste should be documented for record-keeping purposes. In addition to collecting biowaste, the Act gives the possibility of having a composter on private land, which is associated with a partial or total exemption from having a container or bag for bio fraction. In return for the self-treatment of biowaste, the benefit for the residents is the exemption from part of the fees for waste [13]. Using the example of the Jelcz-Laskowice commune in Dolnośląskie Voivodeship, the fee rate for municipal waste management is EUR 6.55 per inhabitant. The fee is reduced by EUR 0.70 per capita for biowaste management with household composters. Such a facility encouraged residents to take the initiative to create their own composter. The result was a reduction in residents' fees and relief from the municipal waste collection company in terms of collecting and processing the bio fraction [14].

Composting is the decomposition process of organic matter under aerobic conditions by bacteria, fungi, and actinomycetes [15]. Microorganisms use the organic substances contained in the waste and transform them into a source of energy and nutrients in the presence of oxygen. Some carbon remains in the biological mass, and the rest is released into the atmosphere as carbon dioxide. Proteins, carbohydrates, and fats are hydrolyzed and converted into organic acids and carbon dioxide. A large amount of heat is released in the process. The entire composting process takes place in four stages, each of which differs in the activity of microorganisms. The first three phases usually last from 3 to 4 weeks and depend on the required quality of the final product [16]. Each phase is essential because the process will stop without it. The first stage is the precomposting in mesophilic conditions. It is characterized by the multiplication of mesophilic bacteria and increased temperature in the reactor to 45 °C. The second phase is known as the intensive composting (thermophilic) phase. The temperature fluctuates in the range of 45–55 °C, but sometimes the temperature falls outside this range. The temperature parameter is significant because above 65 °C, the composting process slows down, and undesirable odors may be produced. Thermophilic organisms multiply rapidly, and the products of this stage are carbon dioxide, water, and ammonia. The next phase is the proper composting, in which the temperature drops to 45 °C and mesophilic bacteria and fungi reform. These microorganisms are transformed into lignin and keratin. The maturation phase takes place last. It is the longest-lasting

stage during which a stable product of composting is produced—hummus and macrofauna appear [12,15–18].

Several important factors influence the quality and time of composting. One of them is the moisture of the load. The initial moisture content must be high and stable during the process. Biowaste is characterized by a high water content which is essential for microorganisms. The pH of the environment in which they live influences microorganisms. Low organic acids are formed when the reaction is acidic, extending the composting process [12,16].

The composting process requires an appropriate ratio of C:N, i.e., organic carbon to nitrogen. Microorganisms need these substances to live but with the proper ratio. The optimal value is in the range of 20–30:1. Below 20:1, odor is released, and the compost quality is reduced. When the C:N ratio exceeds 30:1, the process slows down. The final product should have a ratio of less than 20:1 as the high nitrogen content makes it a more valuable fertilizer [16].

The choice of the composting substrate has an important influence on the entire process, as it determines the porosity structure. Free air spaces allow air to move during the process, giving access to it for microorganisms. It is recommended that the value of free air spaces exceeds 30%, but to such an extent as not to disturb the density of the processed substrate [12,16]. Meeting the appropriate requirements during the composting process allows obtaining a biologically stable product free from pathogens and seeds. Importantly, this process allows the reduction of the batch mass by up to 50% [19]. This is an essential advantage as it helps to solve the space problem of biowaste. It also produces waste heat that can be used [15]. The result of the process is compost, which is a natural, actively biological product. Compost contains organic and mineral particles, which gives it great potential for fertilization. Additionally, after reaching the appropriate moisture content, it can be used as a sorbent to neutralize leakages of petroleum substances [20]. In order to improve the composting process, the factors influencing the decomposition reaction are experimented with. The effect of adding effective microorganisms (EM) to the mixture during the composting of sewage sludge mixed with biowaste was tested. The results were compared to the mixture without the addition of the preparation. Unfortunately, the EM agent did not positively affect the composting performance [21]. Much research was carried out on the aeration of the deposit. The study followed by Sołowska et al. showed that the influence of aeration intensity is of great importance in relation to the first phase of composting and can effectively accelerate the process [12,22].

The structure of the paper is as follows: Section 1 includes information on the global problem of waste management, the composting process, and factors influencing it. Section 2 presents the methodology for measuring the effectiveness of the green waste composting process under hyperbaric conditions including the moisture content, pH value, elemental composition, macronutrients content, the respiratory activity of microorganisms, or loss on ignition. Section 3 contains the research results. Section 4 is a discussion summarizing the research outcomes. Section 5 presents the conclusions resulting from the research and further plans of the research team in this regard.

In analyzing the achievements of scientists to date, it was found in the field of composting that no research team has yet investigated the influence of excess pressure on the course of the composting process. Considering this knowledge gap and the constant trend to find new methods of biowaste management, we undertook to investigate the composting process under hyperbaric conditions.

## 2. Materials and Methods

### 2.1. Properties of Substrates

The input material for the composting process in hyperbaric conditions:

- mowed grass from the green areas of the city of Wrocław, 50 mm long,
- mowed grass from the green areas of the city of Wrocław, 150 mm long.

The grass length was determined based on the literature data [21], which says that the volume of free air spaces should be 25–35% of the composted material. Before the organic matter decomposition process, the substrates were mixed in a 4:1 (a:b) ratio to make them a homogeneous material. Table 1 presents the properties of the substrate subjected to the composting process.

**Table 1.** Properties of the substrate intended for the experiment.

| Parameter | Unit | Mean Value and Standard Deviation |
|---|---|---|
| Bulk density | kg·m$^{-3}$ | 280 |
| Moisture content | % | 60 |
| pH | - | 5.90 |
| Total nitrogen | % | 2.33 (0.21) |
| Total carbon | % | 43 |
| Potassium content | g·kg DM$^{-1}$ | 23.73 (1.26) |
| Phosphorus content | g·kg DM$^{-1}$ | 3.95 (0.02) |
| Loss on ignition | % | 88.97 (0.16) |
| Respiratory activity of microorganisms | mg $O_2$·g DM$^{-1}$ | 109.73 (2.48) |

### 2.2. Description of the Test Stand

The research was carried out in the Institute of Agricultural Engineering at the Wroclaw University of Environmental and Life Sciences.

To carry out the experiment, a test stand was constructed. Figure 1 presents a scheme of the test stand for composting in hyperbaric conditions.

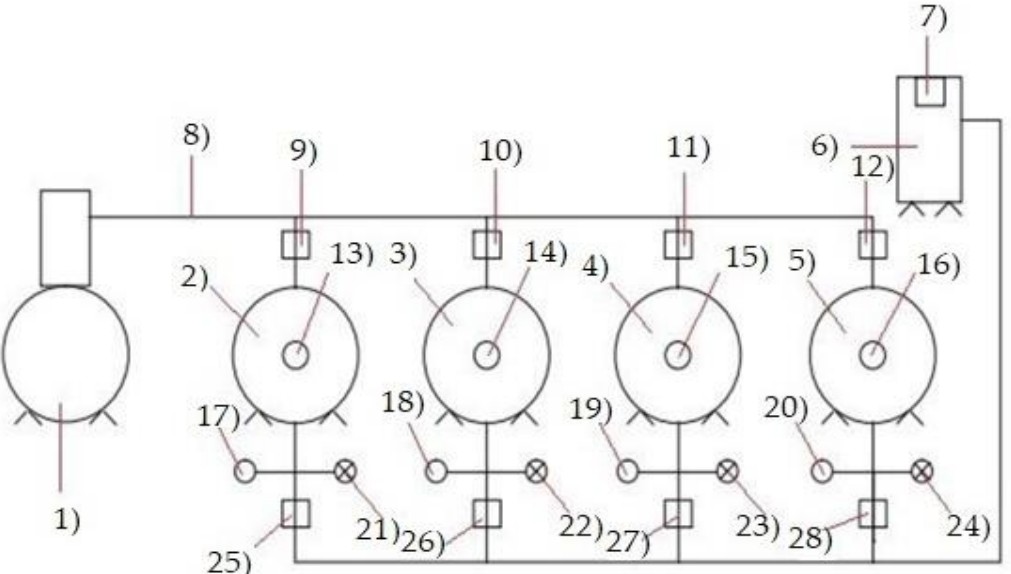

**Figure 1.** Scheme of the test stand for composting under hyperbaric conditions: 1) air compressor; 2), 3), 4), 5) bioreactor; 6) buffer tank for measuring the concentration of carbon dioxide; 7) carbon dioxide sensor; 8) air duct; 9), 10), 11), 12) inlet pneumatic solenoid valve; 13), 14), 15), 16) temperature sensor; 17), 18), 19), 20) overpressure sensor; 21), 22), 23), 24) overpressure gauge; 25), 26), 27), 28) pneumatic output solenoid valve.

Figure 2 presents a general view of the test stand, i.e., reactors, air distribution system and control system.

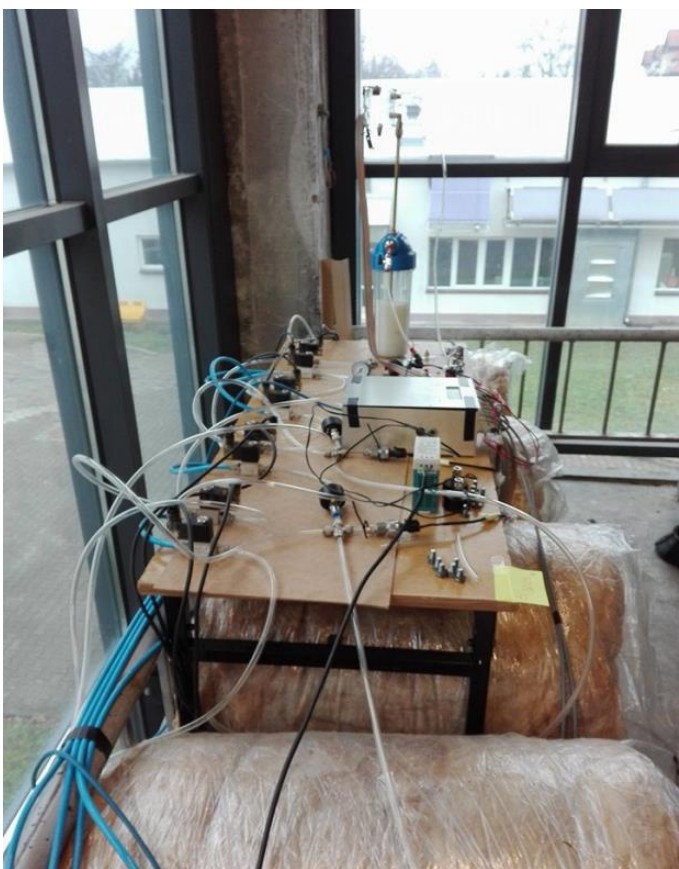

**Figure 2.** General view of the test stand.

The control of the composting process in the bioreactor was possible thanks to the proprietary software of the "BioSerwis" company.

### 2.3. Plan of the Experiment

The experiment plan took into account selected technological parameters:

(a)  air exchange frequencies: 4 and 8 h;
(b)  preset overpressures: 0, 50, 100, and 200 kPa;
(c)  the number of repetitions for each set overpressure and the air exchange frequency: 5.

The process of organic matter oxygen decomposition was carried out in four overpressure variants (0, 50, 100, and 200 kPa) and two air exchange frequencies (4 and 8 h). The duration of a single process was 240 h. As part of the experiment, 40 experiments were carried out (20 in each air exchange frequency). Each variant was performed in 5 repetitions. Samples were prepared on the day the experiment began. Each time, an equal mass of the material (2000 g) was weighed and placed in the bioreactor, with a capacity of 0.05 m$^3$. Then the bioreactors were closed, and the air was forced into them using an air compressor to the assumed overpressure. Pneumatic valves were responsible for maintaining the appropriate overpressure in the bioreactor.

### 2.4. Waste Research Methods

Before and after the composting process under hyperbaric conditions, the parameters presented in Table 2 were determined in the research material.

**Table 2.** Determined parameters in the research material.

| Parameter | Determination Method | Device |
| --- | --- | --- |
| Moisture content | PN-EN 14346: 2011 | PS 3500 R2 laboratory scale, KbC-65W laboratory dryer |
| pH | PN-EN 15011-3:2001 | VOLTCRAFT PHT-200 pH meter |
| Elemental composition | Gas chromatography | CE Instruments CHNS elemental composition analyzer |
| Macronutrients | PN-EN ISO 11885:2009 | ICP-AES iCAP 7400 atomic emission spectrometers |
| Respiratory activity $AT_4$ | OxiTop® Control | OxiTop®-C 110 set, Q-Cell 140/40 laboratory incubator |
| Loss of ignition | PN-EN 15169:2011 | AS 220.R2 laboratory scale, SNOL 8.2/1100 muffle furnace |

*2.5. Statistical Data Analysis*

The results of measuring green waste parameters after 40 composting processes (4 pressures × 2 air exchange frequencies × 5 repetitions) were subjected to statistical analysis, which consisted of the following stages:

- for all measured quantitative variables, basic descriptive statistics were calculated, i.e., mean values (M), standard deviations (SD), medians (Me), lower (Q1) and upper (Q3) quartiles, as well as extreme values Min and Max;
- for all quantitative variables, the compliance of their distribution with the normal distribution was checked using the Shapiro–Wilk test, the homogeneity of variance was checked with the Bartlett and Levene test;
- the analysis of variance for the two-factor classification was used to assess the impact of the independent variables ($p$ and $t_{AE}$) on the dependent variables;
- for all the statistical tests used, the significance level $\alpha = 0.05$ was adopted.

The calculations used the STATISTICA v. 12.5 computer software package and the EXCEL spreadsheet.

In the conducted experiments (composting processes), the independent variables were

- $X_1$—$p$, overpressure in the bioreactor chamber (0, 50, 100, and 200 kPa);
- $X_2$—$t_{AE}$, frequency of air exchange in the bioreactor (4 and 8 h).

The dependent variables were the measurement results:

- $Y_1$—$AT_4$, respiratory activity of microorganisms within 4 days, mg $O_2 \cdot$g $DM^{-1}$,
- $Y_2$—MC, moisture content of the material, %,
- $Y_3$—LOI, loss on dry matter ignition, %,
- $Y_4$—$\Delta m$, loss of dry matter, %,
- $Y_5$—pH value,
- $Y_6$—N, average share of nitrogen, %,
- $Y_7$—C, average share of carbon, %,
- $Y_8$—K, average share of potassium, ‰,
- $Y_9$—P, average share of phosphorus, ‰.

**3. Results**

*3.1. Effectiveness of the Composting Process in Hyperbaric Conditions*

The research on the effectiveness of the composting process took into account

- change in the respiratory activity of microorganisms within 4 days,
- change in material moisture,
- change in organic matter content,
- weight loss of the material subjected to the experiment,
- change in the pH of the composted material,
- change in the macronutrient content of the material.

3.1.1. Degree of Respiratory Stabilization of the Product $AT_4$

The results of the $AT_4$ respiratory activity of biowaste after different variants of the composting process are shown in Figure 3.

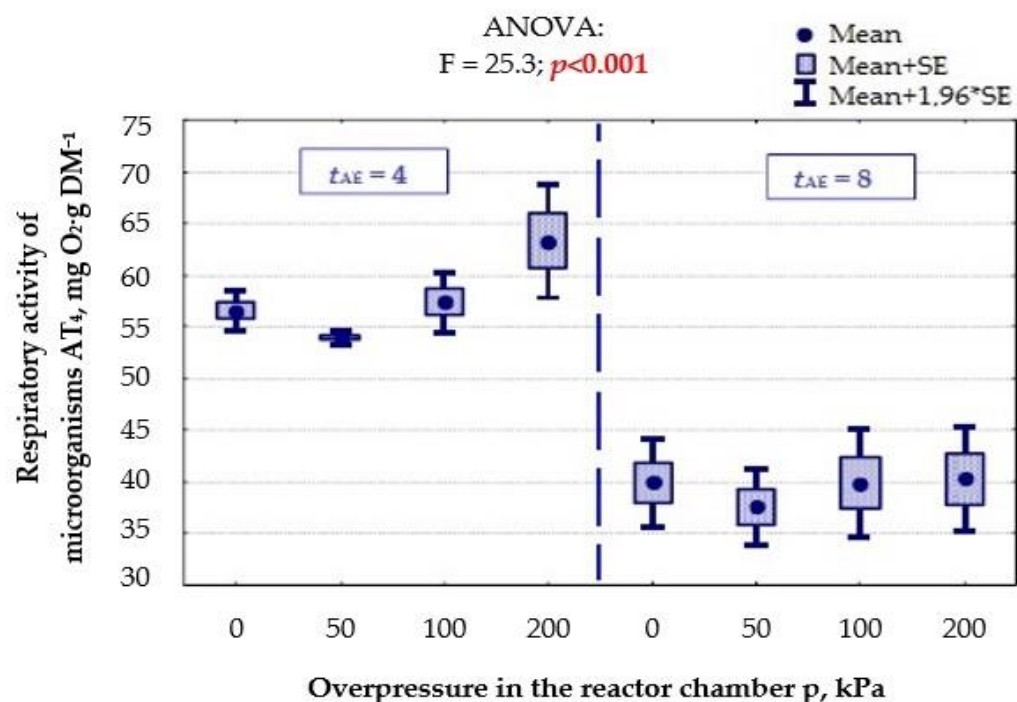

**Figure 3.** Respiratory activity of organisms ($AT_4$) in the bioreactor after the composting process at four overpressures $p$ and two air exchange frequencies $t_{AE}$ and the results of the significance test.

The material subjected to the composting process was characterized by quite high $AT_4$ respiratory activity, amounting to 109.73 mg $O_2 \cdot g \cdot dm^{-1}$. Composting the material under all given conditions resulted in a decrease in the value of the $AT_4$ parameter. During the variant with $t_{AE}$ = 4 h, the highest efficiency in reducing respiratory activity was obtained for overpressure of 50 kPa, where the stabilization efficiency in relation to the parameter of the input material was 50.9%. The variant of composting with an overpressure of 200 kPa turned out to be the least influencing of the respiratory activity reduction. The value of the $AT_4$ parameter in this composting variant decreased by 42% in relation to the parameter of the initial material. During the composting process with the air exchange frequency every 8 h ($t_{AE}$), the highest efficiency of the biowaste mixture stabilization was achieved for variants with an overpressure of 50 and 200 kPa. At an overpressure of 50 kPa, the value of the respiratory activity parameter decreased by 67.5%, and at an overpressure of 200 kPa by 67.6%. The lowest efficiency of material stabilization was recorded for the 100 kPa variant, in which the efficiency of lowering respiratory activity was on the average level of 62.9%.

No statistically significant differences were observed in the effectiveness of the stabilization degree of $AT_4$ in most of the tested overpressures for $t_{AE}$ = 4 h and $t_{AE}$ = 8 h. Statistically, significant differences were only noticeable between the periods of air exchange $t_{AE}$ = 4 h and $t_{AE}$ = 8 h, ($p < 0.01$). The stabilization efficiency was significantly ($p < 0.05$) higher at $t_{AE}$ = 8 h.

### 3.1.2. Moisture Content in the Material

The results of the moisture content in the material after the composting process under various conditions are shown in Figure 4.

The moisture of the input material subjected to the composting process was 60%. During the tests, it was observed that the moisture of the material increased during composting with $t_{AE}$ = 4 h. The highest increase in moisture was observed during the composting process at atmospheric pressure and amounted to an average of 6%. The process carried out at an overpressure of 200 kPa showed a minor influence on the increase in the moisture of the input material. Composting in these conditions increased the moisture content of the material by, on average, 3.3%. In hyperbaric conditions with $t_{AE}$ = 8 h, the moisture of the

material decreased only in the variant with an overpressure of 50 kPa (a decrease of 1.2%). The composting process in the remaining variants increased the material moisture in the range from 0.3% (atmospheric pressure) to 2.8% (overpressure 100 kPa).

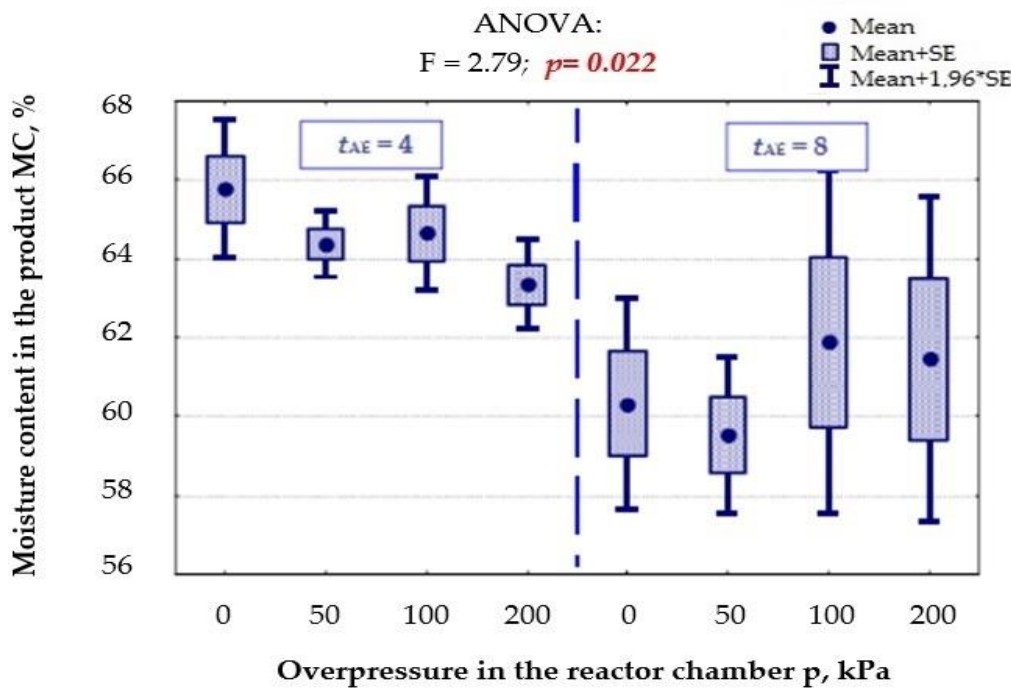

**Figure 4.** Average moisture content in the product after the composting process at four overpressures *p* and two air exchange frequencies $t_{AE}$.

In the test of the significance of the mean relative moisture of the composted material, in most of the tested pressures for $t_{AE}$ = 4 h and $t_{AE}$ = 8 h, no statistically significant differences were observed. Indeed, the statistical difference was only noticeable between periods of air exchange $t_{AE}$ = 4 h, atmospheric pressure (0 kPa), and $t_{AE}$ = 8 h, overpressure 50 kPa ($p < 0.05$).

3.1.3. Loss on Dry Matter Ignition

The results of the loss of organic matter in the composting process (LOI) are shown in Figure 5.

The content of organic matter (LOI) in the material subjected to the composting process in hyperbaric conditions was 88.97%. A decrease in organic matter content in the recycled biowaste was observed at $t_{AE}$ = 4 h in all overpressure variants. The decrease in organic matter content, depending on the variant, ranged between 3.5 and 8.10%. In the results of losses on the ignition after the composting process at $t_{AE}$ = 8 h, higher efficiency of organic matter removal was noted than in the case of $t_{AE}$ = 4 h. The decrease in organic matter content, depending on the variant, ranged from 3.9 to 7.70%. In both types of composting with ventilation of the reactor chamber every 4 and 8 h, the most significant loss of organic matter was recorded for the composting process at an overpressure of 200 kPa.

In the test of the significance of the average effectiveness of losses on the ignition, no statistically significant differences were observed in most of the tested pressures for $t_{AE}$ = 4 h and $t_{AE}$ = 8 h. In fact, the statistical difference was noticeable only between the air exchange frequency $t_{AE}$ = 4 h, overpressure 200 kPa, and $t_{AE}$ = 8 h, atmospheric pressure (0 kPa) ($p < 0.05$), and in the period $t_{AE}$ = 8 h, the loss on ignition was statistically higher at a pressure of 200 kPa than at 0 kPa (atmospheric pressure) ($p < 0.05$).

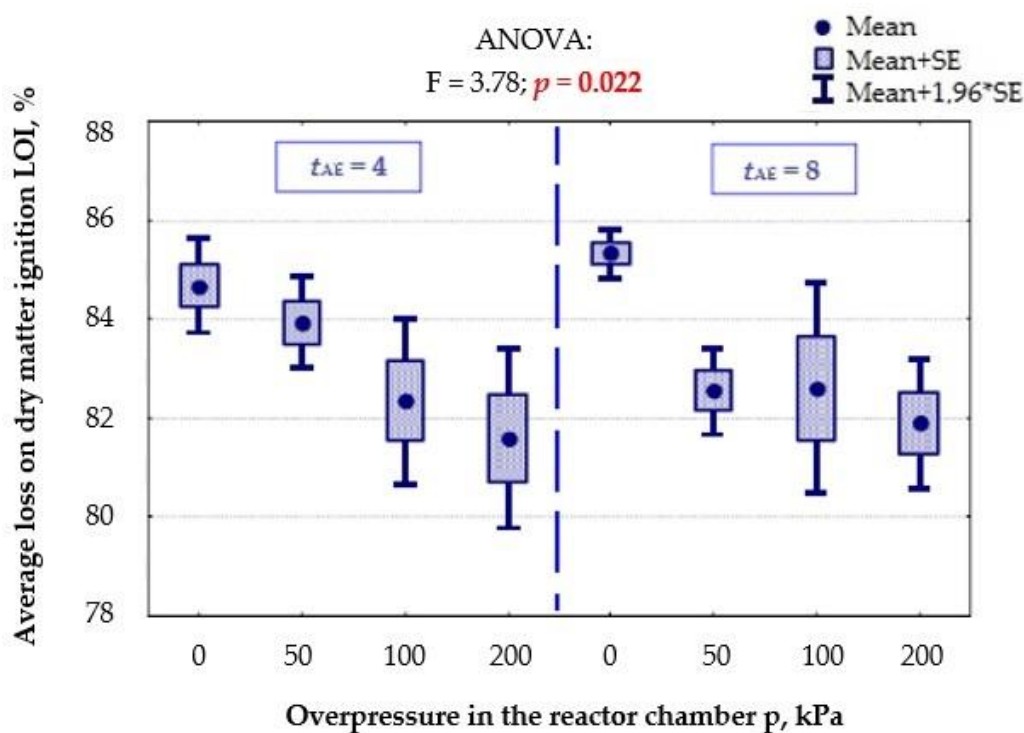

**Figure 5.** Average loss on dry matter ignition (LOI) in the bioreactor after the composting process at four overpressures $p$ and two air exchange frequencies $t_{AE}$.

### 3.1.4. Product Weight Loss

The results of the average loss of specific mass of the composted material are shown in Figure 6.

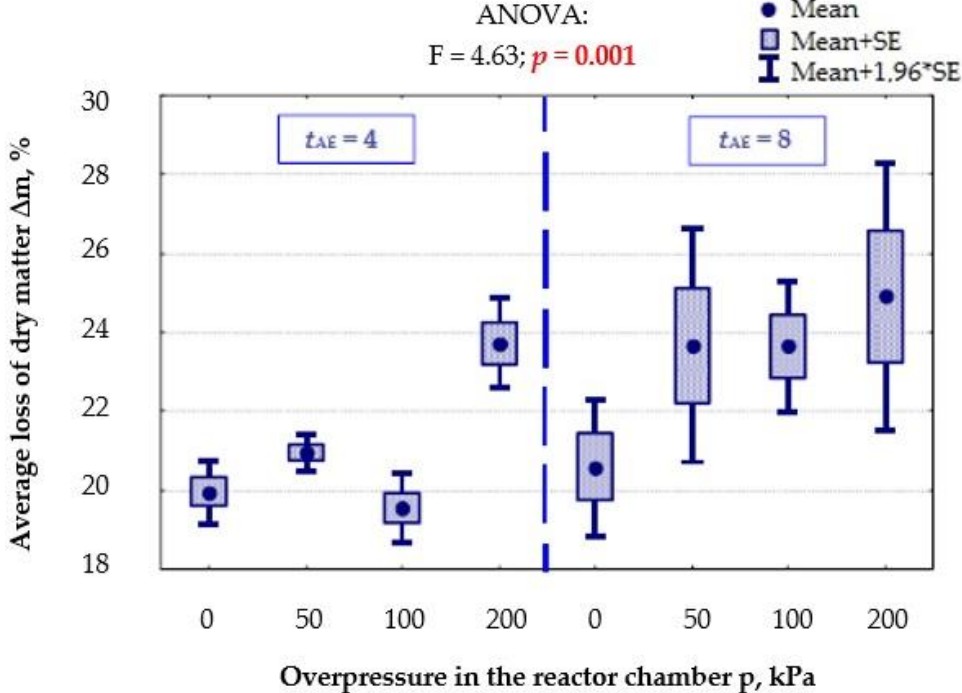

**Figure 6.** Average loss of composted mass ($\Delta m$) in the bioreactor after the composting process at four overpressures $p$ and two air exchange frequencies $t_{AE}$.

After the conducted research, it was observed that with $t_{AE} = 4$ h, the most significant weight loss of the charge material was characteristic for the variant with an overpressure of 200 kPa, which was a weight loss on average of 23.7% compared to the initial weight, which was 2000 g. The lowest degree of weight loss was recorded for composting under 100 kPa overpressure conditions, and it was 19.6%. Comparing the weight loss with $t_{AE} = 8$ h variants, it can be concluded that the overpressure of 200 kPa caused the most significant loss of specific mass, which was on average 25.5% in relation to the initial mass of the charge. The most minor degree of reduction of the initial weight of biowaste was observed for the composting process at atmospheric pressure, and it amounted to 20.8%.

In the test of the significance of the mean weight loss of the composted material in most of the tested pressures for $t_{AE} = 4$ h and $t_{AE} = 8$ h, no statistically significant differences were observed. The statistical difference was noticeable only between the air exchange times $t_{AE} = 4$ h for atmospheric pressure (0 kPa), and 100 kPa as well as for $t_{AE} = 8$ h and overpressure 200 kPa ($p < 0.05$), where the average loss of composted mass was statistically higher in the period $t_{AE} = 8$ h.

3.1.5. pH Value

The results of the pH value of the composted mass are shown in Figure 7.

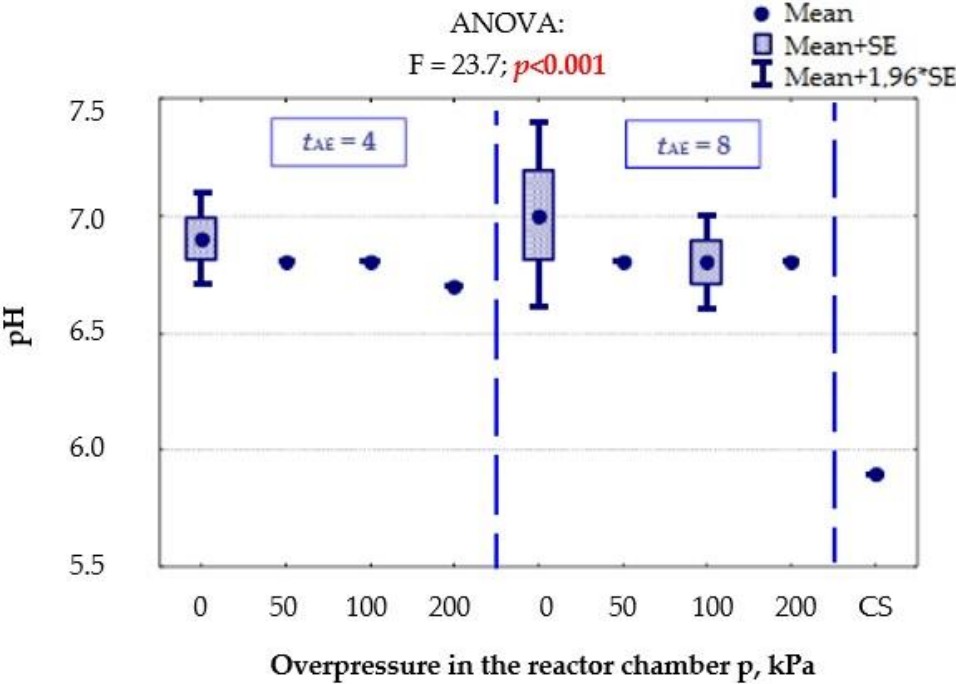

**Figure 7.** The pH of the material after composting for four overpressures $p$, two air exchange frequencies $t_{AE}$, and the control sample CS.

During the composting process, with $t_{AE} = 4$ h, an increase in pH from 5.9 to a maximum of 6.9 was observed in the process carried out in atmospheric pressure (0 kPa). In the remaining overpressure variants (50, 100, and 200 kPa), the pH value increased on average to 6.8. The pH of the composted material, which before the process was 5.9 pH, increased steadily for both periods of air exchange. The most significant change was found in the variant without hypertension. In this case, the pH increased to an average of 7. The remaining pressure variants were characterized by pH values of the composted material close to $t_{AE} = 4$ h, which were on average 6.8.

In the test of the significance of the reaction in all the tested pressures for $t_{AE} = 4$ h and $t_{AE} = 8$ h, no statistically significant differences were observed. A statistically significant difference was noticeable only between the periods of air exchange $t_{AE} = 4$ h, $t_{AE} = 8$ h, and

the control sample ($p < 0.01$). The change in the pH of the composted material in the two assumed periods of air exchange was statistically higher than in the control sample (CS).

### 3.1.6. Elemental Composition Content in the Product

The content of nitrogen element (N) in the material after the composting process is shown in Figure 8.

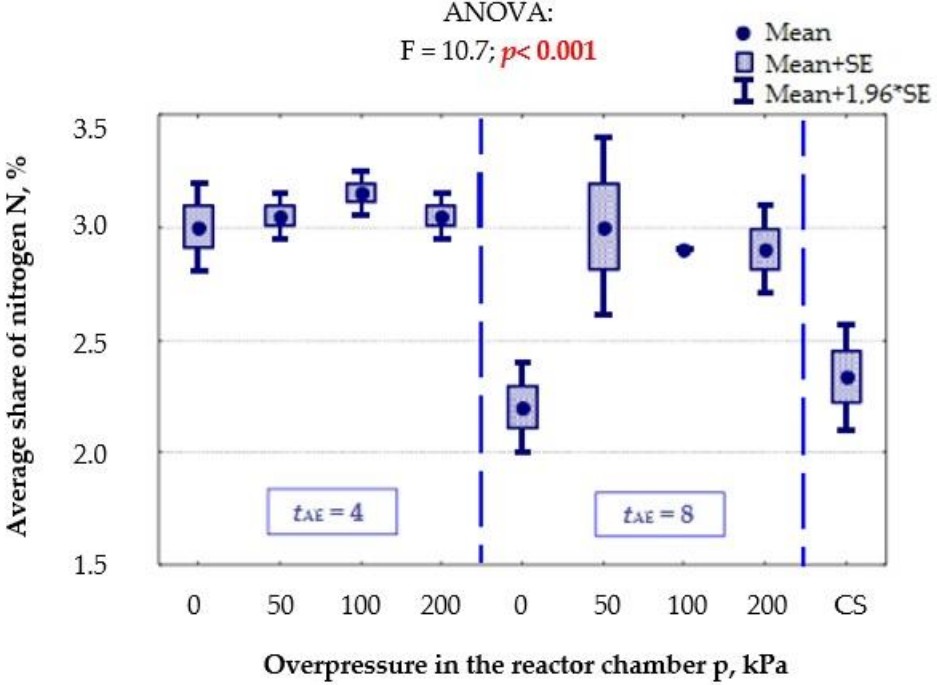

**Figure 8.** Average share of nitrogen (N) in the material after composting for four overpressures *p*, two air exchange frequencies $t_{AE}$, and the control sample CS.

After the composting process was carried out at the air exchange frequency $t_{AE} = 4$ h, we noticed that the overpressure variant 100 kPa had the most significant impact on the N content in the material. The increase in nitrogen concentration in this case was 0.85%. With the air exchange frequency $t_{AE} = 8$ h, the highest increase in the N content was recorded for the variant with an overpressure of 50 kPa, which was 0.70%.

In the test of the significance of the share of nitrogen (N), no statistically significant differences were observed in all the tested pressures for $t_{AE} = 4$ h. Statistically, the significantly lower N content was noticeable at $t_{AE} = 8$ h and atmospheric pressure in relation to all pressures tae = 4 h and other pressure variants at $t_{AE} = 8$ h. The nitrogen content in this test was similar to the share of N in the control sample (CS). Statistically higher content of N occurred at all pressures $t_{AE} = 4$ h and $t_{AE} = 8$ h and an overpressure of 50 kPa than in the control sample ($p < 0.03$).

The content of the carbon element (C) in the composted material is shown in Figure 9.

Due to the decomposition of organic matter, a decrease in the carbon content in the composted material was observed. With the air exchange frequency $t_{AE} = 4$ h, the decrease in carbon concentration was the highest in the variants of atmospheric pressure (0 kPa) and overpressure 100 kPa, the decrease was on average 4.65%. During the composting process in all overpressure variants at the air exchange frequency $t_{AE} = 8$ h, a uniform decrease in the carbon content in the material was noted. The reduction of the carbon content in the material in these variants oscillated at the level of 5.82%.

In the test of the significance of the share of carbon (C), in all the tested pressures for $t_{AE} = 4$ h and $t_{AE} = 8$ h, no statistically significant differences were observed in the control sample (CS).

The content of the element potassium (K) in the composted material is shown in Figure 10.

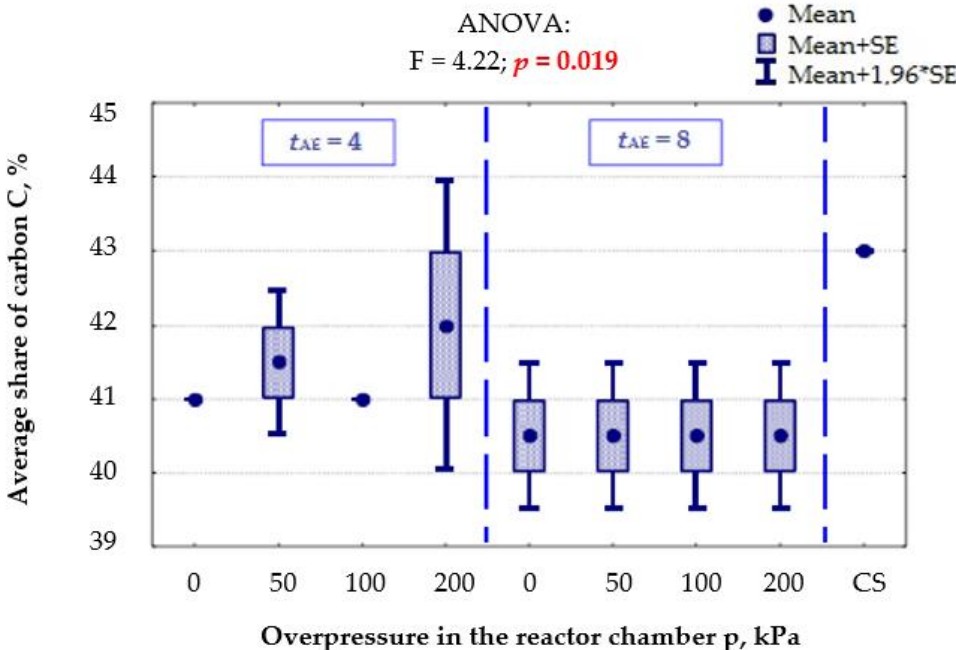

**Figure 9.** Average share of carbon (C) in the product for four overpressures *p*, two air exchange frequencies $t_{AE}$, and the control sample CS.

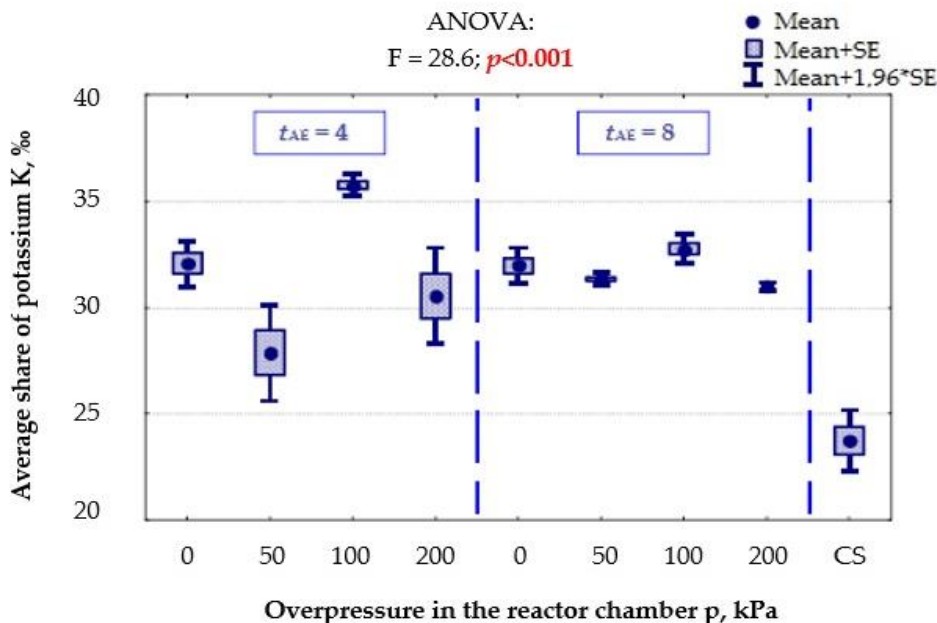

**Figure 10.** Average share of potassium (K) in the product for four overpressures *p*, two air exchange frequencies $t_{AE}$, and the control sample CS.

The highest increase in potassium concentration in the composted material, with the air exchange frequency $t_{AE}$ = 4 h, was recorded for the variant of overpressure of 50 kPa. This increase averaged 14.7%. The highest increase in the potassium content in the composted material with the period of $t_{AE}$ = 8 h was observed for the variant of overpressure of 100 kPa. In this case, the K content increased by 9.1%.

In the test of the significance of the potassium share (K), no statistically significant differences were observed in all the tested pressures for $t_{AE}$ = 8 h. Statistically, significantly higher K contents were noticeable during air exchange every 4 h for all overpressure

variants. The same relationship was observed for $t_{AE}$ = 8 h and overpressure of 50 and 200 kPa. Statistically higher potassium content was in all pressures for $t_{AE}$ = 4 h and $t_{AE}$ = 8 h than in the control sample ($p < 0.04$).

The content of the element phosphorus (P) in the compost is shown in Figure 11.

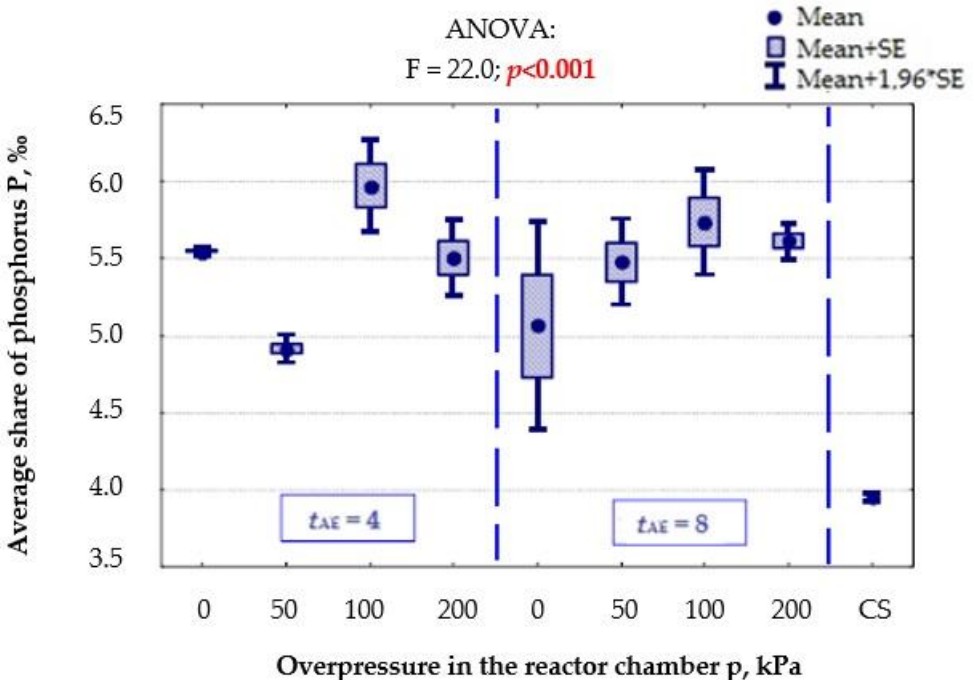

**Figure 11.** Average share of phosphorus (P) in the product for four overpressures *p*, two air exchange frequencies $t_{AE}$, and the control sample CS.

In the case of the content of the element phosphorus (P), an increase in its concentration in the composted material was noted. On average, the content of phosphorus increased by 33.8% for the variant of an overpressure of 100 kPa and $t_{AE}$ = 4 h. Also during composting with the air exchange frequency $t_{AE}$ = 8 h and overpressure of 100 kPa, an increase in phosphorus content in the composted material by an average of 31.2% in relation to the initial material was observed.

Statistically higher content of phosphorus (P) is in all overpressure variants for $t_{AE}$ = 4 h, as well as in all pressure variants for $t_{AE}$ = 8 h, than in the control sample ($p < 0.02$).

## 4. Discussion

Eight experiments, four with air exchange frequency $t_{AE}$ = 4 h and four with air exchange frequency $t_{AE}$ = 8 h, were carried out. Each experiment differed in variants: variant 1—0 kPa (atmospheric pressure) and overpressure variants: 2—50 kPa, 3—100 kPa, 4—200 kPa. In all experiments and variants, tests were carried out on the same input material, characterized by an initial moisture of 60% and a weight of 2000 g. During the composting process of green waste, all parameters of the obtained product were also determined.

The activity of microorganisms during the $AT_4$ test decreased in all tested variants of pressure and ventilation of the bioreactors. The starting value was on average 109.73 mg $O_2 \cdot$g $DM^{-1}$. The highest decrease in the respiratory activity of microorganisms was characteristic for the overpressure variants 50 kPa and 200 kPa at the air exchange frequency $t_{AE}$ = 8 h. There, the respiratory activity of microorganisms decreased to 38.60 mg $O_2 \cdot$g $DM^{-1}$. At the frequency $t_{AE}$ = 4 h, the $AT_4$ value decreased to 53.90 mg $O_2 \cdot$g $DM^{-1}$ in the case of 50 kPa, and it was the lowest value in the bioreactor ventilation frequency. Though not as low as the results of the studies conducted by Lasaridi et al., in which there

was a decrease from the starting value of 55 mg $O_2 \cdot g$ $DM^{-1}$ up to 22 mg $O_2 \cdot g$ $DM^{-1}$ [22]. This value corresponds to stabilized composts [23]. As pointed out by Sidełko et al., and Lornage et al., such decreases are caused by the loss of organic content, which decreases during the process due to mineralization [24,25].

During the research on the moisture of the composted product in hyperbaric conditions with the frequency of ventilation of the composted mass $t_{AE}$ = 4 h, an increase in moisture from 3.3% to 6% was recorded. In the case of the ventilation frequency $t_{AE}$ = 8 h, a decrease in material moisture in relation to the initial value was recorded in the variant of the overpressure of 50 kPa. In the other variants, there was an increase from 0.3% (0 kPa) to 2.8% (100 kPa). It was found that with the ventilation frequency $t_{AE}$ = 8 h, a lower increase in material moisture was obtained. In the case of a decrease in material moisture, it is recommended to water piles of composted material, but a more affordable form is to add water-retaining materials in appropriate proportions, which will retain water in the material [26]. Similarly, Ruggieri et al., recommended irrigation of stockpiles for aerated systems. However, the researchers used a closed composting system without drainage to maintain the moisture at an appropriate level [27].

After analyzing the results, we noticed that the best efficiency of the decrease in organic matter content was characterized by the frequency of ventilating the bioreactors at $t_{AE}$ = 8 h. The efficiency of removing organic matter was higher by 1.6% than the tests at the air frequency of $t_{AE}$ = 4 h. Oxygen losses occurred at $t_{AE}$ = 8 h, during which the anaerobic processes took place. As noted by Ball et al., losses on ignition are related to the aerobic activity of microorganisms. The immature compost samples showed 23 g·kg $LOI^{-1}$, and in the mature compost samples, it was 18 g·kg $LOI^{-1}$ [28]. The input material containing 76 g·kg $LOI^{-1}$ is also presented for comparison. In the case of research on losses on the ignition after composting in hyperbaric conditions, the average loss at the frequency of $t_{AE}$ = 4 h was 85.26%; at the frequency of $t_{AE}$ = 8 h, it was 83.64%; and losses on ignition of the material before composting amounted to 88.97%. Barrena et al. defined compost stability as the degree of degradation of biodegradable materials [29]. When the compost is unstable and therefore immature, as in the studies by Ball et al., where there was a significant difference in the g·kg $LOI^{-1}$ content, it contains significant amounts of biodegradable material that allows the development of highly active microorganisms [28]. However, the research was not aimed at fully stabilizing the organic matter, which would require a much longer hold time. The aim was to demonstrate the effect of applying increased pressure and ventilation intensity on the transformation of organic matter decomposition in the initial phase of the process. Research on the pressure flow and aerobic conditions on the stabilization and maturation of the compost should be continued.

The highest loss of waste mass was characteristic for the variant with an overpressure of 200 kPa at the air exchange frequency $t_{AE}$ = 8 h (25.5% in relation to the initial mass of the substrate intended for the process); similarly, with the air exchange frequency $t_{AE}$ = 4 h, the highest loss was also characterized by variant 200 kPa. The smallest weight loss occurred at $t_{AE}$ = 4 h, 100 kPa, and it amounted to 19.6% in relation to the initial mass of the substrate. The overall reduction in volume, and hence the loss of mass, results from the decomposition of organic matter into other volatile compounds ($CO_2$, $H_2O$, $NH_3$) and from fragmentation and compaction of the material throughout the process [30]. The loss of mass is related to the moisture content of the material after the composting process because the condensed steam that was released during the research process remained at the bottom of the bioreactor.

Bhat et al., reported that the pH is very important in assessing the maturity and quality of the compost. In all tested pressure and aeration variants ($t_{AE}$ = 4 h and $t_{AE}$ = 8 h), an increase in pH from 5.9 was noted [31]. The atmospheric pressure (0 kPa) showed the highest increase in the variant $t_{AE}$ = 8 h. It increased to pH 7; in the remaining pressure variants, the pH was recorded at 6.8. Hogg et al., found that compost in the pH range of 6–8.5 is ideal for soil application [32]. The results of the test of the significance of the content of the elements showed an increase in the content of N in almost all tested pressure variants

at two frequencies of ventilating the bioreactor in relation to the control sample (CS), which is a typical phenomenon related to the removal of carbon, hydrogen, and oxygen as a result of the decomposition of organic matter. Only at the atmospheric pressure (0 kPa) at $t_{AE}$ = 8 h, no differences with the substrate were noticed. According to Cáceres et al., and Nolan et al., nitrification promotes the natural acidification of the compost material as a result of the release of $H^+$ [33,34]. The increase in nitrogen content is related to the decrease in organic carbon content in the composting product. Analogous to the increase in N, the content of C in the composted mixture decreases. As expected, the C/N ratio decreases, also reported by Villaseñor et al. [35]. Additionally, during the experiment, the content of potassium and phosphorus increased in relation to the content in the control sample (CS) in all variants of pressure and frequency of air exchange. Pramanik et al., believed that the increase in phosphorus is due to the formation of acid during the decomposition of organic waste and is responsible for the dissolution of insoluble phosphorus [36]. Similar relationships were shown by the studies conducted by Kaosol et al., where an increase of these two elements was also noted after the composting process [37].

## 5. Conclusions

The approach presented in this manuscript is novel. So far, no research team has undertaken to investigate the effect of hyperbaric conditions on the quality of produced compost.

The analysis of the test results allows to confirm that composting in hyperbaric conditions causes

(a) the highest weight losses of the product at the overpressure variant of 200 kPa ($t_{AE}$ = 4 h—23.7% and $t_{AE}$ = 8 h—25.5%).
(b) the highest efficiency of organic matter removal for overpressure variants 100 and 200 kPa ($t_{AE}$ = 4 h), and for overpressure variants 50, 100, and 200 kPa ($t_{AE}$ = 8 h).

It allows to conclude that hyperbaric conditions during composting increase the efficiency of the process.

The overpressure composting method presented in this article is an innovative approach to this process. The method proposed by our team allows for a more effective conversion of green waste to compost. This research makes it possible to implement this type of solution in urban greenery companies. They can use this method and produce valuable fertilizer that they can either use on their own or sell for additional income.

Depending on the aeration conditions, traditional composting methods take 6 to 8 weeks. After composting in hyperbaric conditions, this process was shortened to an average of 10 days. This paper also creates a space for further research on the impact of the use of catalytic additives on the quality of the produced compost or the creation of the process energy balance. The research team also plans to continue the tests on a laboratory scale by narrowing down the overpressure variants from 100 kPa to 200 kPa, taking into account the air exchange intervals that would not create anaerobic conditions.

**Author Contributions:** Conceptualization, K.R. and J.B.; methodology, K.R.; software, K.R.; validation, J.B.; formal analysis, K.R.; investigation, K.R.; resources, K.R.; data curation, K.R.; writing—original draft preparation, B.G., B.K. and S.G.; writing—review and editing, B.G. and B.K.; visualization, K.R.; supervision, J.B. All authors have read and agreed to the published version of the manuscript.

**Funding:** The APC is financed by Wroclaw University of Environmental and Life Sciences.

**Institutional Review Board Statement:** Not applicable.

**Informed Consent Statement:** Not applicable.

**Data Availability Statement:** The data presented in this study are available on request from the corresponding author.

**Conflicts of Interest:** The authors declare no conflict of interest.

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
