# Peer review of "Analysis of the Effectiveness of Green Waste Composting under Hyperbaric Conditions"

_sustainability, doi:10.3390/su14095108_

Round 1

Reviewer 1 Report

Overall, this paper is very readable, the structure is fairly clear, the methods are well described, the results are detailed presented, and the discussion is sufficient. However, this paper can still be improved:
Abstract: the authors should focus more on the main conclusion (results) instead of the introduction and methods.
Introduction: the literature is not sufficiently reviewed, and the research gap this study filled is not clearly stated.
Table 2 is not necessary since the content in it has been described in the sentences above.
The authors should pay more attention to the details, such as the significant number (e.g., line 295, line 300).

Author Response

  1. Abstract: the authors should focus more on the main conclusion (results) instead of the introduction and methods.

As suggested by reviewer, more obtained results has been presented in the abstract

  1. Introduction: the literature is not sufficiently reviewed, and the research gap this study filled is not clearly stated.

The research gap of this study has been added to the introduction part.

  1. Table 2 is not necessary since the content in it has been described in the sentences above.

Table 2 has been deleted from the manuscript as suggested by reviewer.

Reviewer 2 Report

The paper present interesting research with the useful application.

I have no comments except small mistake below.

--------

Maybe please correct the Figures 9, 10 description:

Figure 9. Avarage share of carbon (C) -   AVERAGE !

Author Response

  1. Maybe please correct the Figures 9, and 10 description.

As the reviewer suggested, description of Figures 9 and 10 has been corrected.

Reviewer 3 Report

  • Add research data and main implication in the abstract.
  • Identify the gap in research in introduction. Please add.
  • What is the state-of-the-art for composting process under hyperbaric conditions? Add some details.
  • Are 10 days sufficient for compost stabilization. Have you checked degradation index?

Author Response

  1. Add research data and main implication in the abstract.

As suggested by reviewer, more obtained results have been presented in the abstract.

  1. Identify the gap in research in introduction. Please add.

The gap in research in the introduction has been identified.

  1. What is the state-of-the-art for composting process under hyperbaric conditions? Add some details.

As suggested reviewer, some details presenting the main advantage of novel composting under hyperbaric conditions has been added to the text. In further experiments, it is planned to couple the test stand (composting reactors) with a set of electric heaters in order to maintain optimal thermal conditions of the process.

  1. Are 10 days sufficient for compost stabilization. Have you checked degradation index?

The degradation index was not used during the tests. In order to determine the stabilization of the compost, the following indicators were used: C/N ratio, pH, AT4. After periodic control of the above-mentioned parameters, the average composting time was determined at the level of 10 days. The assessment of compost stability varies between EU countries. In Poland, compost stabilization is determined on the basis of the above-mentioned parameters.

Round 2

Reviewer 3 Report

The manuscript is improved and can be accepted for publication.